# Dynamical strengthening of covalent and non-covalent molecular interactions by nuclear quantum effects at finite temperature

Huziel E. Sauceda [1,2,3✉], Valentin Vassilev-Galindo [1], Stefan Chmiela [2], Klaus-Robert Müller [2,4,5,6✉] & Alexandre Tkatchenko [1✉]

Nuclear quantum effects (NQE) tend to generate delocalized molecular dynamics due to the inclusion of the zero point energy and its coupling with the anharmonicities in interatomic interactions. Here, we present evidence that NQE often enhance electronic interactions and, in turn, can result in dynamical molecular stabilization at finite temperature. The underlying physical mechanism promoted by NQE depends on the particular interaction under consideration. First, the effective reduction of interatomic distances between functional groups within a molecule can enhance the $n \rightarrow \pi^*$ interaction by increasing the overlap between molecular orbitals or by strengthening electrostatic interactions between neighboring charge densities. Second, NQE can localize methyl rotors by temporarily changing molecular bond orders and leading to the emergence of localized transient rotor states. Third, for noncovalent van der Waals interactions the strengthening comes from the increase of the polarizability given the expanded average interatomic distances induced by NQE. The implications of these boosted interactions include counterintuitive hydroxyl–hydroxyl bonding, hindered methyl rotor dynamics, and molecular stiffening which generates smoother free-energy surfaces. Our findings yield new insights into the versatile role of nuclear quantum fluctuations in molecules and materials.

[1] Department of Physics and Materials Science, University of Luxembourg, L-1511 Luxembourg City, Luxembourg. [2] Machine Learning Group, Technische Universität Berlin, 10587 Berlin, Germany. [3] BASLEARN, BASF-TU joint Lab, Technische Universität Berlin, 10587 Berlin, Germany. [4] Department of Artificial Intelligence, Korea University, Anam-dong, Seongbuk-gu, Seoul 02841, Korea. [5] Max Planck Institute for Informatics, Stuhlsatzenhausweg, 66123 Saarbrücken, Germany. [6] Google Research, Brain team, Berlin, Germany. ✉email: sauceda@tu-berlin.de; klaus-robert.mueller@tu-berlin.de; alexandre.tkatchenko@uni.lu

Nuclear delocalization is a fundamental feature of quantum mechanics resulting from Heisenberg's uncertainty principle. In molecules, light elements such as protons and first row atoms are especially prone to delocalization. Even molecules or materials with heavier atoms and strong bonds can exhibit significant nuclear quantum effects (NQE)[1–6]. Within the Born-Oppenheimer (BO) approximation, NQE tend to lower energy barriers and stimulate tunneling. In addition, the inclusion of NQE promotes a delocalized sampling of the molecular configuration space, consequently exploring regions of the potential-energy surface (PES) inaccessible by classical dynamics. As a result, this can enhance or inhibit certain molecular interactions[7]. A clear example is the hydrogen bond, where the NQE affect interactions in biological systems and molecular crystals by delocalizing protons. In the case of bulk water, NQE can even qualitatively change its fundamental physical and chemical properties[8,9]. In general, the study of NQE in molecular and biological systems is a thriving research field covering from rigid and fluxional molecules[10–13] to liquids and DNA base pairs[14–19], allowing the analysis of, for example, interactions between neighboring molecules via hydrogen bonding[7,8,20], spectroscopic properties[21,22] and proton transport[23]. Nevertheless, previous works have been mainly focused on the general implications of proton delocalization and much less is known about how NQE influence other types of covalent and non-covalent interactions. Particularly, biological systems often use combinations of covalent and non-covalent interactions for carrying out a wide variety of different processes. Therefore, it is crucial to understand whether NQE can also play an important role beyond hydrogen bonding.

In this work we report counterintuitive effects induced by NQE: nuclear delocalization can lead to a dynamical strengthening of covalent and non-covalent molecular interactions at finite temperature. Our conclusions are shown to be valid for a wide range of molecules. We have found that the explicit mechanism responsible of enhancing each interaction depends on its underlying physical or chemical origin. For example, NQE-induced increase in interatomic distances can modify the local environment of methyl rotors affecting their dynamics. At the same time, NQE increase the molecular polarizability, enhancing the dispersion interactions between molecules or molecular fragments. In other cases, NQE can shorten distances between molecular fragments, thereby boosting interactions that depend on such distances. In order to demonstrate these results, we have selected representative mechanisms ubiquitously occurring in biological systems: $n \rightarrow \pi^*$ interactions, methyl rotors, as well as electrostatic and van der Waals interactions. The faithful description of such weak molecular interactions require high levels of theory (e.g., coupled cluster) which is not always computationally affordable when performing long ab initio path integral molecular dynamics (PIMD) simulations. In this study, we have performed PIMD simulations using machine learned molecular force fields constructed using the sGDML framework[11,21,24–28] and trained on coupled cluster reference data [CCSD(T) or CCSD depending on the size of the molecule] (See Supplementary Note 1 for more details).

## Results

We have chosen to study the role of NQE in a series of small molecules that serve as fundamental examples of mechanisms that are present in larger chemical and biological systems. We start by analyzing $n \rightarrow \pi^*$ interactions in aspirin as an example of local electronic orbital effects influenced by nuclear quantum delocalization. We then proceed to study the unexpected NQE-induced localization of methyl rotor in toluene as a model

for methyl groups in biomolecules. Finally, we analyze the strengthening of electrostatic interactions in paracetamol and van der Waals interactions in different conformations of benzene dimer.

To do so, we employ the symmetric gradient domain machine learning (sGDML) framework to reconstruct the molecular force fields[11,21]. This approach makes full use of physical symmetries (i.e. permutational invariance and energy conservation) as priors, which enables particularly data-efficient models. In previous work, we have demonstrated that sGDML yields accurate descriptions of quantum effects in small flexible molecules[25]. Performing predictive calculations require a series of stringent demands on the setting of the simulations, such as highly accurate description of the quantum interactions, extensive sampling of the potential energy surface, as well as the incorporation of the nuclear quantum fluctuations. Thereby, in this article we have used coupled-cluster reference data as well as accurate density-functional approximations to train the sGDML molecular force fields and the NQE were included via the path integral molecular dynamics formalism. The molecular dynamics simulations were performed at room temperature using the i-PI package and its interface to the sGDML force field[29], and all the simulations presented here were done for at least 500 ps with time steps of 0.2 fs. Additional post-processing ab initio calculations were done using methods such as natural bond orbital (NBO)[30] to compute the $n \rightarrow \pi^*$ interactions and symmetry-adapted perturbation theory (SAPT)[31] to study the non-covalent intermolecular interactions. The SAPT and coupled cluster calculations were done using the Psi4 package[32], while the NBO calculations we done using the ORCA package[33]. See Methodology section and Supplementary Notes 1 and 3 for an extended description of the methods used in this work.

**Enhanced $n \rightarrow \pi^*$ interaction.** A particularly important type of interaction often occurring between pairs of neighboring carbonyl groups is the so-called $n \rightarrow \pi^*$ interaction. It arises from the delocalization of lone–pair electrons on electronegative atoms (e.g., oxygen atom) into an antibonding $\pi^*$ orbital of an aromatic ring or a carbonyl group (see Fig. 1A)[34]. First discussed in early 1970, the $n \rightarrow \pi^*$ interaction has attracted significant attention in recent years and it is hypothesized to impart substantial structural stability to proteins[35–38] and molecules[39–42], as well as define reactivity[40], regulate isomerisation[34] and energy barriers[43], and promote charge transfer[44]. Nevertheless, the actual dynamical implications at finite temperature of such interaction have not been explicitly studied.

To elucidate this matter, here we study the aspirin molecule as a proof of concept. For this molecule the $n \rightarrow \pi^*$ interaction is the main contribution to the relative energy of the global minimum (Fig. 2a) and two other local minima (Fig. 2b, c), thereby defining their energetic ordering[39,40]. Figure 2 shows the configuration space sampling obtained from classical MD and PIMD simulations at room temperature, where the dynamical implications of the NQE on aspirin's behavior are evident: NQE constrains the dynamics of the molecule to the global minimum in contrast to the results from classical MD. Hence, the NQE must be promoting a particular intramolecular interaction and, given the evidence provided by Choudhary et al.[40], the $n \rightarrow \pi^*$ interaction between the ester and carboxyl groups is the main candidate. To investigate further the contribution of the $n \rightarrow \pi^*$ interaction to the total energy, we have computed the $n \rightarrow \pi^*$ interaction energy $E_{n \rightarrow \pi^*}$ along the ester's minimum energy pathway (MEP) trajectory (Fig. 1A) using NBO analysis[30,45] (see section Methods for computational details). In what follows, we use the NBO definition for the $n \rightarrow \pi^*$ energy[30,45], whereby a

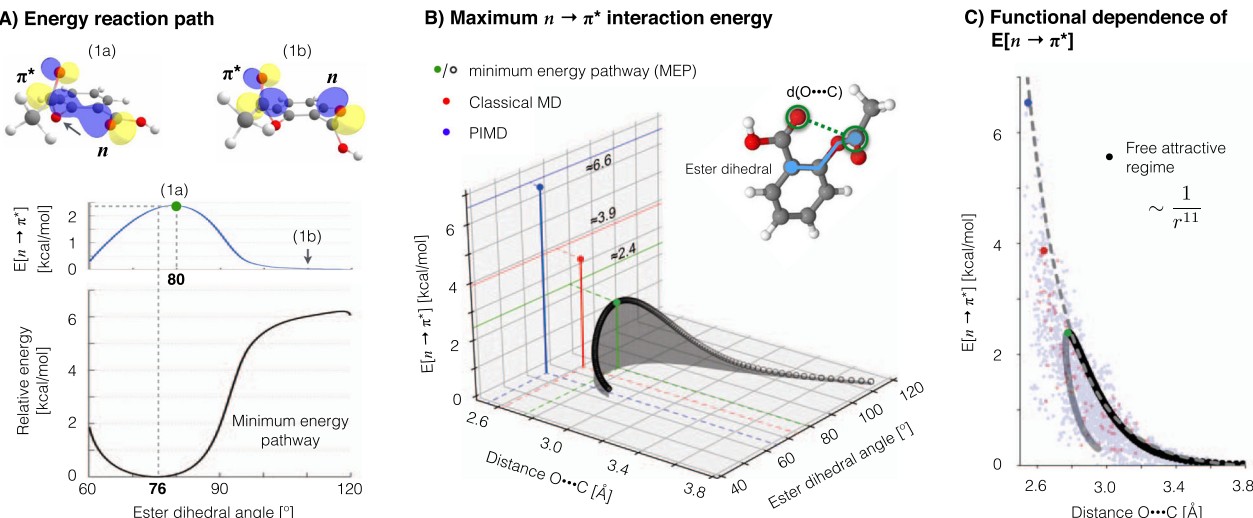

**Fig. 1 Enhancement of the $n \to \pi^*$ interaction by nuclear delocalization in Aspirin. A**-Bottom Aspirin's potential energy surface and **A**-Middle $n \to \pi^*$ interaction energy $E_{n \to \pi^*}$ along the minimum energy pathway (MEP) of the ester's dihedral angle. The energy $E_{n \to \pi^*}$ was computed using natural bond orbital (NBO) method[30,45] at CCSD/cc-pVDZ level of theory, whereby a positive energy value means stabilization and zero energy means absence of overlap between $n$ and $\pi^*$ orbitals. Hence, a positive value implies a stabilization of the molecule. Both plots are in the same energy scale. **A**-Top The configuration (1a) (marked by the green circle) defines the maximum interaction (stabilization) energy $E_{n \to \pi^*}$ along the MEP, while (1b) represents a configuration where overlap between lone-pair electrons $\phi_{(n)}$ and the antibonding $\phi_{(\pi^*)}$ orbitals, and therefore the energy $E_{n \to \pi^*}$, has gone to zero. **B** Estimations of the maximum $E_{n \to \pi^*}$ interaction energy values reached while running PIMD (blue circle, $\approx 6.6$ kcal mol$^{-1}$) and classical MD (red circle, $\approx 3.9$ kcal mol$^{-1}$) at 300 K using the sGDML@CCSD model. As a reference, the energy $E_{n \to \pi^*}$ curve is plotted along the MEP trajectory as a function of its two main degrees of freedom, the interatomic distance $d_{O \cdots C}$ and the ester's dihedral angle. The maximum energy $E_{n \to \pi^*}$ value along the MEP is $\approx 2.4$ kcal mol$^{-1}$ (green circle). **C** Approximate functional dependence of the interaction energy $E_{n \to \pi^*}$ on the oxygen (in hydroxyl) and carbon (in ester) interatomic distance $d_{O \cdots C}$. The $1/r^n$ function was fitted to the free-attractive-regime part of $E_{n \to \pi^*}$ (black circles) along the MEP starting from 3.8 Å, giving a value of $n \sim 11$.

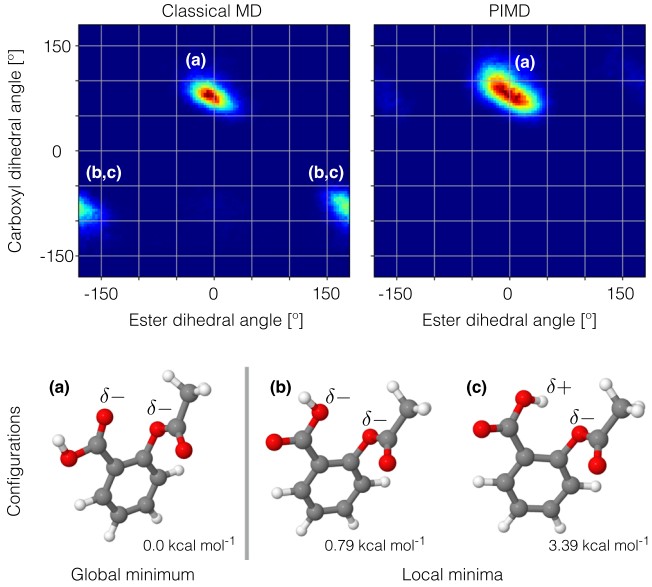

**Fig. 2 Classical (MD) and path integral molecular dynamics (PIMD) simulations at room temperature of aspirin described by the sGDML@CCSD molecular force field.** The plots are projections of the dynamics to the two main degrees of freedom of aspirin: carboxyl and ester dihedral angles. Structures of three relevant molecular configurations are shown: **a** global minimum and **b**, **c** two of the lowest local minima.

positive energy value means stabilization and zero energy means absence of overlap between $n$ and $\pi^*$ orbitals. Hence, a positive value of the $n \to \pi^*$ energy in Fig. 1 implies stabilization of the molecule. The visual representation of the $n$ electron delocalization into $\pi^*$ is presented in Fig. 1-A-Top. These results show that

the $E_{n \to \pi^*}$ is maximum near the global minimum of aspirin (green circle in Fig. 1A-Middle, 1a), such value quickly vanishes as the molecule moves away towards the transition state (at 180°). From these results we see that the $n \to \pi^*$ interaction contributes ~40% of the global energy minimum relative to the transition state. Additionally, we also found that the two main degrees of freedom describing $E_{n \to \pi^*}$ in aspirin are the $d_{O \cdots C}$ distance, also known as Bürgi-Dunitz parameter[34,35,37,40,41], and the ester's dihedral angle, as shown in Fig. 1B.

Now, in order to understand the results from finite temperature simulations (Fig. 2) in the context of the $n \to \pi^*$ interaction (Fig. 1A-Middle), we have computed the $E_{n \to \pi^*}$ energy for a set of configurations sampled from the classical MD and PIMD trajectories on the global minimum (Fig. 2a), and then we have plotted their maximum $E_{n \to \pi^*}$ respective values in Fig. 1-B. All the computed samples are plotted against the Bürgi-Dunitz parameter $d_{O \cdots C}$ in Fig. 1C.

The results displayed in Fig. 1B already provide a clear picture of the behavior of the $n \to \pi^*$ interaction at finite temperatures: The maximum $E_{n \to \pi^*}$ energy along the MEP (i.e. 0K) is of 2.4 kcal mol$^{-1}$ (green circle in Fig. 1), but this value can be enhanced by 160% due to pure thermal fluctuations (red circle in Fig. 1) and up to 270% by NQE at room temperature (blue circle in Fig. 1). This means that the NQE alone could strengthen the attractive interaction energy between the carbonyl and the ester functional groups by up to ~2.7 kcal mol$^{-1}$ at room temperature. Consequently, given the evidence of such a considerable increment of the $n \to \pi^*$ interaction energy and the configurational localization resulting from the molecular dynamics simulations, both originated by the NQE, we have found that nuclear quantum delocalization can stabilize intramolecular interactions and selected molecular conformations.

It is worth to analyze the underlying dynamics created by NQE that lead to such a prominent increase of the $E_{n \to \pi^*}$ energy, which

could suggest ways to generalize the results found here to other systems. From the MEP trajectory in Fig. 1B we can see the approximate dependence of the $E_{n\rightarrow\pi^*}$ energy as a function of the $d_{O\cdots C}$ distance and the ester's dihedral angle. Given the nature of the $n \rightarrow \pi^*$ interaction, i.e., its increase with the orbital overlap[38], variations of the $d_{O\cdots C}$ distance should generate the steepest changes of $E_{n\rightarrow\pi^*}$. This can be seen in Fig. 1B, where a small decrease of the $d_{O\cdots C}$ and dihedral values increase the interaction energy. More interestingly, if we only focus on the free-attractive-regime of the interaction energy as displayed in Fig. 1C, e.g., qualitatively described by $E_{n\rightarrow\pi^*}(r) \sim \int \phi^*_{(n)}(r-x)\phi_{(\pi^*)}(x)dx$ with $\phi_{(X)}$ being the respective molecular orbitals shown in Fig. 1A-Top, the fitting of a $r^{-n}$ function to the attractive part of the MEP trajectory suggests that the $n \rightarrow \pi^*$ interaction energy can be approximated by $E_{n\rightarrow\pi^*}(r) \sim r^{-11}$ (dashed line in Fig. 1C). This approximation serves as a upper limit envelope to the out-of-equilibrium configurations sampled from classical MD and PIMD simulations (red and blue circles in Fig. 1, respectively), and even extrapolates to the more extreme cases such as the maximum energy value reached by the quantum dynamics. Such a steep dependence on the distance between functional groups reveals that even a minor nuclear quantum delocalization leads to a substantial increase in stability.

From these results, and based on the fact that $n \rightarrow \pi^*$ interactions have been consistently reported to occur in different molecular and biological systems[34–40,40–44], we can hypothesize that the strengthening of such interaction by the NQE at finite temperature could prompt similar localization effects in biological systems. Hence, we conclude that nuclear quantum fluctuations are not only the source of the enhanced sampling in atomic systems, but also they can promote molecular and inter-molecular rigidity in systems with prominent $n \rightarrow \pi^*$ interactions

such as polyproline helices in protein fragments which displays a double carbonyl–carbonyl interaction[41].

**Methyl rotor hindering**. The methyl (Me) functional group is a pervasive fragment in chemical and biological systems, playing a fundamental role in, for example, genetics[46] and protein synthesis[47]. The immediate chemical neighborhood of the Me group can drastically modify its energy landscape, going from a free rotor to a localized one with large energetic rotational barriers.

In general, NQE are known to play an important role in lowering energetic barriers when these are of the order of $k_BT$. In the particular case of the Me group, rotational barriers can be much lower than $k_BT$. For this reason the Me group is often considered to be a nearly free rotor at room temperature ($k_BT \approx$ 0.6 kcal mol$^{-1}$ at $T = 300$ K). The toluene molecule is one of the simplest representative examples of a molecule with a Me group. The Me rotor in toluene has a sixfold rotational PES whose best experimental estimates for the energetic barrier range from $\approx 0.014$ to 0.028 kcal mol$^{-1}$ ($\approx 4.9$–9.8 cm$^{-1}$)[48–50], while theoretical results at the CCSD(T) level of theory give $\approx 0.024$ kcal mol$^{-1}$ (blue curve in Fig. 3E). After performing MD simulations at room temperature (classical MD and PIMD) at the sGDML@CCSD(T) level of theory and analyzing the Me rotor's dynamics, we found contrasting results to what can be trivially assumed given the nature of the system. One would expect that NQE lower the rotational energy barriers even more, but Fig. 3B shows that NQE actually hinder the Me group rotations (red) contrary to the free rotation obtained from classical MD (blue). In fact, an incremental inclusion of the NQE via increasing the number of beads in PIMD simulations demonstrates that nuclear delocalization systematically localizes the Me rotor dynamics. Additionally, the PIMD results show that the Me group no longer

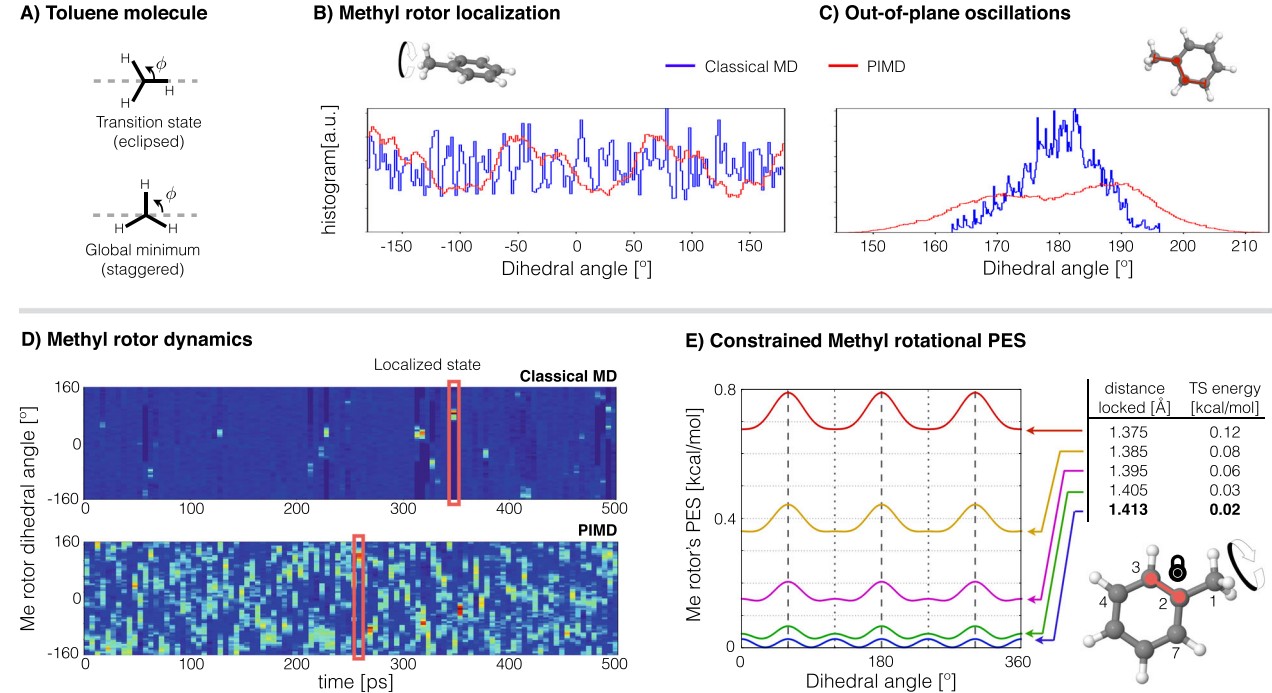

**Fig. 3 Hindering of methyl rotor dynamics by nuclear delocalization in toluene molecule. A** Global minima and transition state of Me rotor. Histogram of the Me rotor's dihedral angle (**B**) and out-of-plane dihedral angle (**C**) computed from classical MD (blue) and PIMD (red). **D** Time evolution of the Me rotor for classical MD (top) and PIMD (bottom) simulations. The red rectangles indicate some of the localized states in the dynamics. The size of the bins are 5 ps × 6°. The simulations were done at room temperature at the sGDML@CCSD(T) level of theory. **E** Methyl rotor's PES and its transition state (TS) energy for different fixed values of the $C_2 - C_3$ distance (marked in red). In a localized state, there is a quasi-linear correlation between $C_2 - C_3$ and $C_2 - C_7$ distances, $d_{C_2 - C_7} \sim -d_{C_2 - C_3}$ (See Supplementary Note 7 for further details).

stays in the plane defined by the benzene ring as in the classical case, instead higher amplitude out-of-plane oscillations are observed due to the NQE (Fig. 3C).

To understand the origin of this localization, we first focus on the time evolution of the Me rotor shown in Fig. 3D. Here we can see that the classical description of the rotor is indeed a free rotor most of the time, nevertheless an interesting phenomenon emerges: The Me rotor can suddenly stop rotating for up to 4 ps. Still this is not apparent from the cumulative histograms in Fig. 3B, C due to the rare nature of this event. Contrasting with the classical model, PIMD results show a qualitatively different picture. In this case the rotor localization is much more frequent but in general the lifetime of the localized state is shorter. From this, we can hypothesize that nuclear quantum delocalization promotes the localization of the Me rotor, but at the same time the NQE tunnel the system out of the localized state. In contrast, Me rotor localization is a rare event in classical dynamics, but when it occurs, it can take a longer time for purely thermal fluctuations to bring the system out of such state. An important remark is that the Me rotor hindering mechanism in toluene is a general phenomenon independent of the employed level of electronic structure theory. We found similar results for Hartree-Fock and density-functional calculations at the PBE0 and PBE0 +MBD[51,52] levels of theory even though their rotational energy barriers are only ~0.001, 0.008 and 0.009 kcal mol$^{-1}$, respectively (See Supplementary Section 6 and Supplementary Fig. 2).

The hindering of the rotations of the Me group has a dynamical origin, and it is determined by the delocalization of the benzene carbon-carbon bonds generated by the NQE. Bond length delocalization is a well known implication of NQE, which, in this particular case, transforms the Me rotor's PES from a six-fold energy surface to a three-fold energy surface as shown in Fig. 3E. Furthermore, the magnitude of the transition state (TS) energy is determined by the two benzene ring bonds $C_2 - C_3$ and $C_2 - C_7$ near to the Me group (see Fig. 3A, E). Consequently, PIMD will tend to generate much higher rotation energy barriers given the extra dilation of such bond lengths induced by the NQE beyond the thermal dilation generated by classical MD (See also Supplementary Fig. 3). According to our results, the rotor described by PIMD experiences energetic barriers of up to 0.55 kcal mol$^{-1}$, energy comparable to $k_B T$, therefore hindering Me rotations. From here we can conclude that the intricate quantum dynamics exhibited by the Me rotor in toluene is due to two competing NQE: On one side the nuclear delocalization of the carbon atoms promotes higher rotational energetic barriers hindering the rotor, but the quantum fluctuation of the hydrogen atoms in the Me takes the rotor out of the localized state.

Even though the results shown here are for toluene, the electronic origin of the rotational energetic barrier of Me rotor is very similar in different molecular systems (See Supplementary Fig. 3)[50,53–55]. Hence, similar dynamical effects are to be expected in large biological systems given the ubiquity of methyl groups in macro-molecules and protein fragments.

**Electrostatic interaction strengthening**. Up to now, we have shown examples of how NQE enhance covalent interactions, either by local perturbations of the electronic structure or by promoting electronic transitions. Now we turn to study how NQE can affect electrostatic interactions between charged fragments or molecules. For example, it could happen that nuclear delocalization would lead to a decrease in the distance between functional groups with opposite effective charges, thereby strengthening their direct electrostatic attractive interaction. In general, electrostatic interactions are known to play a fundamental role in biological systems given their long range effects. At the

intramolecular level, the interaction between effective atomic charges is weak compared to covalent forces. Nevertheless, the non-bonded fragments within the same molecule can substantially affect each other due to the closeness between them.

Here, we illustrate this effect by analyzing the case of the paracetamol molecule. Figure 4A shows the charge–charge interaction between the oxygen atom in the acetamide functional group and one of the hydrogen atoms in the benzene ring. By performing classical MD and PIMD simulations at room temperature, we found that the interatomic distance between the two atoms $d_{O \cdots H}$ and the acetamide's main dihedral angle $\phi_{Ace}$ display correlated dynamics (Fig. 4B). Effectively, the surfaces displayed in Fig. 4B represent the free energies of the two simulations, clearly showing that the inclusion of NQE considerably reshapes the free energy surface. The NQE localize both the $d_{O \cdots H}$ and $\phi_{Ace}$ degrees of freedom, and therefore, the average configuration of the molecule changes. Carefully examining the rest of the molecule's degrees of freedom during the dynamics, we see that $d_{O \cdots H}$ is the one that changes the most, mainly due to the proton delocalization to regions closer to the oxygen atom (see Fig. 4C). Therefore, such delocalization of $d_{O \cdots H}$ and the strong localization of $\phi_{Ace}$ observed in PIMD results compared to their weak coupling in classical MD, leads us to conclude that the dynamical localization is due to the strengthened electrostatic interaction between the partially charged atoms, $O^{\delta-} \cdots H^{\delta+}$. Similarly to the $n \rightarrow \pi^*$ interaction where the reduction of the interatomic distance increases the $E_{n \rightarrow \pi^*}$ energy, here the electrostatic energy is also increased, thereby effectively stabilizing the molecule.

**Impact of NQE on van der Waals interactions**. The previous examples dealt with intramolecular interactions. Now we proceed to analyze the potential role of NQE on van der Waals dispersion interactions. There is a plethora of literature on the importance of van der Waals forces in nature, from small molecules to water solvation of proteins[56] and from materials science to cohesion in asteroids[57]. As a result, it is natural to wonder if such omnipresent interaction can be affected by NQE. By considering a simple model for vdW energies, we can see that there is an immediate relationship between them. Starting from the well known $R^{-6}$ approximation to the dispersion interaction energy between two non-bonded fragments A and B: $E_{vdW} \sim C_6^{A,B} R_{AB}^{-6}$. The $C_6^{A,B}$ coefficient is computed via the Casimir-Polder integral, $C_6^{A,B} = 3/\pi \int_0^\infty d\omega \alpha_A(i\omega) \alpha_B(i\omega)$, where $\alpha(i\omega)$ is the frequency-dependent polarizability evaluated at imaginary frequencies[58]. In the case of atoms, it was recently demonstrated that the dipole polarizability is given by $\alpha \sim R_{vdW}^7$ where $R_{vdW}$ is the vdW radius[59]. This implies that subtle increments of the molecular volume driven by temperature or NQE will considerably increase the polarizability, and consequently strengthen vdW interactions. In this regard, Aguado et al.[60] and Sharipov et al.[61] reported studies where they estimated measurable corrections to polarizability by zero-point vibrations for sodium clusters and polyatomic molecules, respectively. In consequence, NQE should have an inevitable impact on dispersion forces according to the Casimir-Polder formula.

As a proof of concept, we have considered here the benzene dimer in its three most studied conformations (see Fig. 5). But before analysing their intermolecular interaction, we test our hypothesis that NQE increase the molecular polarizability. To do so, we have computed the relative increment on the benzene isotropic polarizability compared to the optimized molecular structure. We created two ensembles of 50 molecular configurations, each sampled from classical MD and PIMD simulations (see Supporting Information for more details) and then computed

**A) Electrostatic attraction**

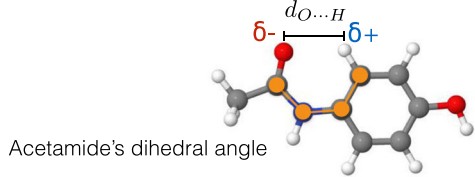

Acetamide's dihedral angle

**B) Acetamide group localization**

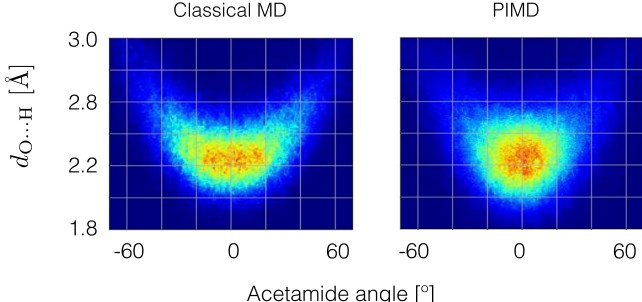

**C) Hydrogen atom delocalization**

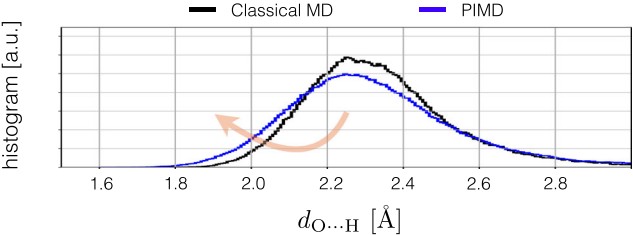

**Fig. 4 Strengthening of electrostatic interactions in paracetamol molecule by proton delocalization. A** Graphical representation of the paracetamol molecule. **B** Effective free energy surface generated from classical MD and PIMD at room temperature using the sGDML@DFT(PBE0 +MBD) force field. **C** Interatomic distance distribution $d_{O\cdots H}$ generated from classical MD (black) and PIMD (blue).

the polarizability tensor using CCSD/aug-cc-pVDZ calculations. Such level of theory recovers 87% of the experimental molecular polarizability for the benzene molecule. Given that here we are interested in analyzing the relative gain in the polarizability by the thermal fluctuations and NQE relative to the equilibrium molecular configuration, such semi-quantitative calculations are sufficient. From this, we obtained that the increment on the mean isotropic polarizability $\langle \frac{1}{3}\mathrm{Tr}(\alpha)\rangle_{ensemble}$ is of ~1% and ~3.4% for the classical and quantum dynamics, respectively, which indeed proves our initial hypothesis.

Now, we proceed to investigate the implications of such NQE-induced increase of the molecular polarizability on noncovalent intermolecular interactions. In order to approximate the inter-molecular interaction between the benzene pair at finite temperature (300 K), we have performed a noncovalent energy surface scanning calculation using molecular configurations sampled from classical MD and PIMD simulations. The interaction energy was computed using symmetry-adapted perturbation theory (SAPT)[31] as implemented in Psi4[32]. Each one of the points in the curves displayed in Fig. 5 is the result of an ensemble average over 50 molecular pair configurations. See Supplementary Fig. 4 for more details on how the dataset was generated. As a reference, we have also performed a normal scanning of the benzene dimer inter-molecular interaction energy (zero Kelvin) to see what is the gain in the interaction energy due to thermal and nuclear quantum fluctuations (labeled as *Relaxed*

in Fig. 5). The cases of the methane dimer and methane–benzene pair are also analyzed in the Supplementary section 8.

From all the cases considered in this study, our main result is the consistent trend displayed by the inclusion of nuclear quantum fluctuations in enhancing the total interaction energy relative to the classical simulations. The results in Fig. 5 show a very revealing behavior which can help understand further the experimental findings for the benzene dimer fluxional dynamics. First, we compare the well known relaxed-configurations energy curves for this dimer. The shape of the curves and energetic ordering of the structures agrees with previously reported results, and in particular the almost degenerate energy values for the parallel-displaced and T-shaped configurations[62]. If we consider the thermal dilation of the molecules, we see that the interaction energy increases as expected, but it is only the inclusion of the nuclear quantum fluctuations which reveals the trends in the benzene–benzene interaction at finite temperature. From Fig. 5 we see that NQE at finite temperature break the energetic quasi-degeneracy between the parallel-displaced and T-shaped config-urations by increasing the energy of the parallel-displaced structure by ~10% relative to the T-shaped one. This suggests that the parallel-displaced benzene dimer configuration would be slightly favored at room temperature. Still it would be expected that the most sampled region of the configuration space is the intermediate state between the two minima as reported by the experiment[63]. In fact, we notice that the T-shaped arrangement of the dimer barely profits (energetically) from the thermal or nuclear quantum fluctuations, contrary to the case of the parallel configuration which is the one that benefits the most (~20%).

From the results presented here, we can conclude that indeed the NQE increase the molecular polarizability and therefore enhance noncovalent interactions. Furthermore, these results suggest that NQE could have a strong impact on large dispersion-dominated biological systems such as proteins and solvated molecules due to their cumulative nature.

Here we studied a model where the benzene molecules are maintained at a fixed distance from each other. While the presented model might be considered idealized, such situations are abundant in nature when the mutual distance between polarizable moieties is effectively fixed by intramolecular constraints or external conditions such as pressure or by confinement of molecules in nanostructures.

## Discussion
The results presented in this study show convincing evidence of intra- and inter-molecular interactions enhanced by NQE, which then promote localized dynamics in molecular configuration space. This finding is counterintuitive, given that it is commonly assumed that NQE tend to delocalize molecular dynamics by expanding covalent bond lengths, increasing non-covalent dis-tances or lowering energetic barriers. Nevertheless, the effects of NQE delocalization are far from being trivial in complex mole-cular conformations that are stabilized by an interplay between covalent and non-covalent interactions. These interactions can conspire to reshape the molecular bond length distribution and the free energy landscape. Even relatively small changes in bond lengths can, in a cumulative manner, bring together non-bonded fragments of the molecules altering their local environ-ments and facilitating the occurrence of unexpected processes. The underlying mechanism responsible for the NQE-induced localization varies depending on the particular interaction. Localization can happen by breaking molecular symmetry as in the case of the Me rotor via increasing the total potential energy and creating a transient well-like potential which confines its dynamics (see Fig. 3E). Alternatively, in the case of non-bonded

**Benzene dimer noncovalent interaction**

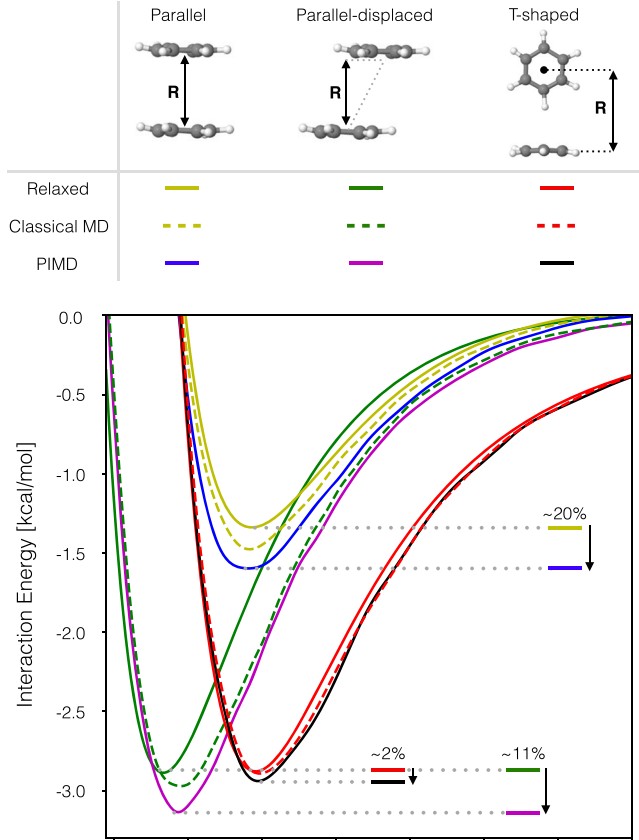

**Fig. 5 Strengthening of noncovalent interactions in benzene dimer by NQE.** Interaction energy calculations scan along the inter-molecular distance $R$ for the parallel, parallel-displaced and T-shaped configurations of the benzene dimer. Each of these curves was calculated for the relaxed configurations and for configurations sampled from classical MD and PIMD trajectories of single benzene molecule. The total noncovalent interaction energies were computed using SAPT0/jul-cc-pVDZ[31] as implemented in Psi4 package[32]. The molecular configurations were sampled from classical MD and PIMD trajectories computed using i-PI software coupled with the sGDML@CCSD(T) force field. See Supplementary Note 8 for more details.

functional groups, NQE can decrease the distance between neighboring fragments thereby promoting orbital overlap as in the case of $n \rightarrow \pi^*$ interaction or increase the electrostatic energy between partial charges, which indeed drastically lowers the energy of the system. On the other hand, the NQE-induced dilation of the molecular effective volume and polarizability increase is the underlying mechanism behind the strengthening of noncovalent van der Waals interactions.

It is clear that in large and complex molecules (for example biological systems) there are many more interactions than the ones discussed here, but the diverse selection of interactions and processes in this study represents several important classes of phenomena that are commonly present in biological systems. A valid argument in this regard is the fact that many of these interactions are weak compared to covalent forces or even hydrogen bonds. Interestingly, this assumption is mostly based on theoretical calculations where the finite temperature effects are ignored. More importantly, here we have shown that nuclear quantum fluctuations can contribute considerably to the enhancement of these seemingly weak interactions. In the case of

dispersion interactions, it is worth to highlight that here we have found that they can be considerably strengthened by NQE. This result suggests that we could have been underestimating the true contribution of vdW forces, given their ubiquity and paramount role in the description of chemical and biological mechanisms.

It is worth to remark that even for efficient machine-learning models, it is not yet computationally feasible to describe real biological systems with tens of thousands of atoms while preserving the accuracy needed to describe many of the subtle quantum effects. Thereby a proper understanding of NQE and their implications on covalent and noncovalent molecular interactions in biologically relevant fragments (e.g., amino acids, DNA base pairs, peptides and small proteins) could be the key to design atomistic molecular force fields capable of accurately capturing complex electronic and nuclear quantum-mechanical effects at finite temperature.

## Methods

**Machine learned force fields: sGDML**. The molecular interactions used for the simulations performed in this work were generated with the symmetric Gradient-Domain Machine Learning (sGDML) framework[11,21]. The sGDML method has been used to obtain accurate reconstructions of flexible molecular force fields from reference datasets of high-level ab initio calculations[25]. More specifically, the different levels of generalization accuracies and applicability of the sGDML framework have been assessed by (1) reproducing the exact trajectories from electronic structure-based MD simulations starting from the same initial conditions[26], (2) recovering the interatomic distance distribution for several molecules relative to ab initio simulations[11,26], and (3) by corroborating experimental molecular vibrational spectra and the statistical occupation values of molecular conformers [21].

Unlike traditional classical force fields, the sGDML approach imposes no hypothesized interaction pattern for the atoms and can thus model any complex molecular interactions[25,64]. Instead, sGDML imposes energy conservation as inductive bias, a fundamental property of closed classical and quantum mechanical systems that does not limit generalization. This allows highly data efficient reconstruction of molecular force fields, without sacrificing generality.

There have been many attempts to accurately learn molecular force fields by using mainly neural networks[65–69] and kernel-based models[70–77], cf. also refs. [78] and [79]. Nevertheless, the sGDML framework offers many advantages desirable when performing highly accurate PIMD simulations, such as high data efficiency during training for reaching state-of-the-art prediction accuracies and computational efficiency[24]. Therefore, to fully reach the limits of the sGDML models' accuracy, an appropriate number of training data points has to be used. In particular, sGDML models display a fast saturation of the generalization accuracy with the size of the training set[21]. In this work, we have used models trained on 1000 molecular configurations, clearly reaching the saturation limit[21]. The generalization error across the potential energy surface for all the molecules included in this study is below 0.2 kcal/mol for energies and below 0.7 kcal mol$^{-1}$ Å$^{-1}$ for forces [24,25].

**Reference molecular databases**. For this study we have used a variety of quantum chemistry levels of theory. The selection of the reference level of theory for each molecule in this study was done based on its accuracy relative to CCSD(T)/cc-pVDZ method. The results for aspirin were generated with models trained on CCSD calculations using the Dunning's correlation-consistent basis set cc-pVDZ, while in the cases of toluene and benzene, we have used CCSD(T)/cc-pVDZ data directly. It is important to highlight that for the small molecules considered in this work, the convergence of their thermodynamic and statistical properties do not require large basis sets, and in particular for the above mentioned molecules this means that their intramolecular interactions are already well captured with cc-pVDZ basis set. As an example, we have performed classical MD and PIMD simulations using a sGDML model for aspirin trained on CCSD/cc-pVTZ reference calculations, results that display the same behavior as the ones reported in Fig. 2 generated using a smaller cc-pVDZ basis set (See Supplementary Figure 1 for a comparison). In the case of the paracetamol molecule, we have used DFT at the PBE0 level of theory with many body dispersion (MBD) method to account for van der Waals interactions[51] using the *really_tight* basis set settings as implemented in the FHI-aims code[80]. Such reference level of theory was used given that PBE0 +MBD yields better relative energies of equilibrium configurations with respect to CCSD(T)/cc-pVDZ compared to CCSD/cc-pVDZ for this specific molecule. See Supplementary Table 1 for a comparison of the energies for the different levels of theory. For further details regarding the accuracy of the *really_tight* basis set refer to the Supplementary Note 2.

To confirm the robustness of our conclusions, we also carried out a range of classical MD and PIMD simulations using sGDML models trained on different levels of theory, such as Hartree-Fock/aug-cc-pVQZ and PBE0+MBD/*really_tight* reference calculations for toluene and aspirin. These additional reference

calculations are free from any basis set artifacts and are converged to meV-level accuracy. Overall, PIMD simulations with these additional sGDML models fully support the results and conclusions presented in the main manuscript. The generated results and their description can be found in the Supplementary Figs. 1 and 2.

All the datasets used in this work were generated by the representative sampling method as reported in ref. [26], page 284.

**Calculation of $E_{n \to \pi^*}$.** Natural Bond Orbital (NBO) second order perturbative energies $E_{n \to \pi^*}$ obtained from NBO 7.0[45] calculations coupled with ORCA 4.1.2[33] at CCSD/cc-pVDZ level of theory were taken as the stabilization energies due to $n \to \pi^*$ interactions. Such interactions, which play an important role in molecular reactivity and conformation (for instance, the Bürgi-Dunitz trajectory[81] preferred during nucleophilic attacks at a carbonyl carbon), comprise delocalization of lone-pair electrons ($n$) of an electronegative atom into an empty $\pi^*$–antibonding orbital of an aromatic ring or a carbonyl group[40,42,82].

## Data availability

All datasets used in this work are available at http://www.sgdml.org or http://quantum-machine.org/datasets/. Additional data related to this paper may be requested from the authors.

## Code availability

The full documentation of the sGDML software can be found at http://quantum-machine.org/gdml/doc/and the code can be downloaded from https://github.com/stefanch/sGDML.

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

## Acknowledgements

V.V.G. and A.T. were supported by the Luxembourg National Research Fund (DTU PRIDE MASSENA) and by the European Research Council (ERC-CoG BeStMo). K.R.M. was supported in part by the Institute of Information & Communications Technology Planning & Evaluation (IITP) grant funded by the Korea Government (No. 2019-0-00079, Artificial Intelligence Graduate School Program, Korea University), and was partly supported by the German Ministry for Education and Research (BMBF) under Grants 01IS14013A-E, 01GQ1115, 01GQ0850, 01IS18025A and 01IS18037A; the German Research Foundation (DFG) under Grant Math+, EXC 2046/1, Project ID 390685689.

## Author contributions

H.E.S. and A.T. conceived the research and designed the analyses. H.E.S and V.V.G. performed quantum chemical calculations. S.C., H.E.S., A.T., and K.R.M. developed the machine learning methodology. H.E.S. performed the molecular dynamics simulations. H.E.S. created the figures with help from other authors. H.E.S., K.R.M. and A.T. wrote the paper. H.E.S., V.V.G., S.C., K.R.M., and A.T. discussed results and commented on the manuscript.

## Funding

## Competing interests

The authors declare no competing interests.
