## [Peer Review File · Nature Communications]

REVIEWER COMMENTS

Reviewer #1 (Remarks to the Author):

The present manuscript contains a numerical study on the impact of nuclear quantum effects (NQE) of covalent and non-covalent interactions. Most importantly they find that NQE tend to generally increase the strength of both covalent and non-covalent interactions. All simulations are technically throughout conducted at the highest possible level. Particularly noteworthy are the usage of force fields trained by symmetric gradient domain machine learning framework based on accurate coupled cluster reference data. In the light that the existing previous studies along very similar lines have been conducted using empirical force fields of density functional theory only, I consider this even the strongest point of the present work.

As such, all of the present calculations and conclusions drawn are reliable and will stand the test of time.

However, as just indicated, there exists quite some previous papers what is even called „competition of NQE“, which renders the degree of novelty of the present work to be average at best. Previous works along those lines include the pioneering work of Markland and coworkers [J. Chem. Phys. 131, 024501 (2009)], Tim Clark et al. [ChemPhysChem 20, 2461 (2019)], as well as Ref. 9 of the present work. If the authors would have not only considered but contrasted the former two works, the authors wouldn't have called their main results „counterintuitive“, but a direct consequence of the well-known competition of NQE, which can play out either way.

To summarize, this is a very fine paper that can be published essentially as is. It is therefore a though call, but given the fact that the first paper claiming the existence of the subtle competition of NQE is now more than 10 years old, the degree of novelty is simply too low to justify publication in Nature Communications.

Reviewer #2 (Remarks to the Author):

The authors aim to elucidate phenomena surrounding the quantum nuclear effect (NQE) in small molecules using the sGDML machine learning-based force field along with the i-PI package to conduct classical molecular dynamics (MD) and path integral molecular dynamics (PIMD). They evaluate three phenomena that are found to be strengthened by NQE, and result in the stabilization of dynamics: the n- π^* interaction in Aspirin, a methyl rotor, and non-covalent interactions. Comparisons between classical molecular dynamics and path integral molecular dynamics simulations using the sGDML model provide evidence for the authors conclusions. The conclusions result in the new insight that NQE not only cause delocalization but also localization in specific cases, which could have relevance in biochemical applications.

I have no specific complaints or observations. Overall, the conclusions in the present article are sound if one trusts the underlying model used to conduct the MD and PIMD simulations. Previously published literature show the sGDML model (data set construction and model itself) can accurately fit to small data sets of small molecules. They have been shown to produce models with energy errors in the range of 0.1-0.3 kcal/mol. Data sets for the specific molecules studied in this work were generated for previous work and evaluated extensively for room temperature dynamics. Therefore, I believe the conclusions of this article to be scientifically sound.

The use of machine learning methods to obtain new insights into chemistry and physics phenomena is intriguing and of interest to a broad audience. This, in my opinion, makes the article a good selection for publication in Nature Communications.

I recommend the article for publication in Nature Communications.

Reviewer #3 (Remarks to the Author):

The manuscript of Saucedo et al. presents a thorough examination of the counter-intuitive contribution of nuclear quantum effects (NQE) on non-covalent interactions, both at intra- and

intermolecular level. This unusual increase is a result of nuclear delocalization, as it was found from ab initio path integral molecular dynamics. I cannot express an opinion on this article since I am not convinced if this conclusion is based on computational artifacts or not.

The authors should provide a numerical argument that the counter-intuitive contribution of NQE is not an artifact of the force fields (FFs) generated by sGDML. Will classical FFs or FFs generated by other ML methods provide the same results? This argument will further enhance their conclusions.

In the Methodology section, the authors mention that the sGDML models were trained on CCSD/cc-pVDZ, CCSD(T)/cc-pVDZ and PBE0+MDB/really_tight. I have many questions here.

1. Which quantum chemical method was used for each molecular system that is discussed on the manuscript?
2. It is well-known that coupled-cluster needs very large basis sets (at least of quadruple-zeta quality or extrapolation to the complete basis set limit) in order to provide the expected accuracy. How the choice of a basis set will affect the PIMD results and, especially, the counter-intuitive phenomena? The cc-pVDZ basis set is an extremely small basis and provides inaccurate results for non-covalent interactions. For example, for the benzene dimer (parallel displacement), a test case that is used by the authors on this paper, the slightly larger aug-cc-pVDZ basis has an error of -1.87 and 0.63 kcal/mol for CCSD and CCSD(T) methods, respectively (see http://vergil.chemistry.gatech.edu/active_bfdb/bfdb/cgi-bin/reactionviewer.py?dataset=S22&rxn=S22-11).
3. What means really_tight for PBE0+MDB. The PBE0+MDB does not require standard basis functions?

Similarly, how accurate are the computed polarizabilities at the CCSD/aug-cc-pVDZ level of theory?

Caption of Figure 3 (S1), please change "inividual" to "individual".

Response to the reviews:

Reviewer #1 (Remarks to the Author):

The present manuscript contains a numerical study on the impact of nuclear quantum effects (NQE) of covalent and non-covalent interactions. Most importantly they find that NQE tend to generally increase the strength of both covalent and non-covalent interactions. All simulations are technically throughout conducted at the highest possible level. Particularly noteworthy are the usage of force fields trained by symmetric gradient domain machine learning framework based on accurate coupled cluster reference data. In the light that the existing previous studies along very similar lines have been conducted using empirical force fields of density functional theory only, I consider this even the strongest point of the present work.

As such, all of the present calculations and conclusions drawn are reliable and will stand the test of time. However, as just indicated, there exists quite some previous papers what is even called „competition of NQE“, which renders the degree of novelty of the present work to be average at best. Previous works along those lines include the pioneering work of Markland and coworkers [J. Chem. Phys. 131, 024501 (2009)], Tim Clark et al. [ChemPhysChem 20, 2461 (2019)], as well as Ref. 9 of the present work. If the authors would have not only considered but contrasted the former two works, the authors wouldn't have called their main results „counterintuitive“, but a direct consequence of the well-known competition of NQE, which can play out either way.

To summarize, this is a very fine paper that can be published essentially as is. It is therefore a though call, but given the fact that the first paper claiming the existence of the subtle competition of NQE is now more than 10 years old, the degree of novelty is simply too low to justify publication in Nature Communications.

Response:

We thank the referee for providing comments on our manuscript and highlighting the reliability of our conclusions and their benchmark nature that “will stand the test of time”, according to the referee. We also thank the referee for mentioning two additional landmark references, which we have cited in the revised manuscript.

As mentioned by the reviewer, previous results have been reported regarding the role of NQE in rigid molecules, mostly water and a few other systems (DNA base pairs, proton transport). These studies have assessed the role of NQE in interactions between neighboring molecules, almost always involving hydrogen atoms or protons. Hence, our study is not the first to highlight the importance of NQE, and we reworded parts of the manuscript such as the introduction by thoroughly acknowledging the state-of-the-art reference works in this field.

Nevertheless, our study goes far beyond current knowledge since we demonstrate the implications of NQE on a much wider range of interactions in flexible molecules covering intra- and inter-molecular interactions and a wealth of different electronic mechanisms, which leads to key findings of great importance for biology and molecular chemistry. Furthermore, we show that the physical mechanism behind NQE depends on the type of interaction itself, which then highlights

the intricate coupling between the energy landscape anharmonicities and the nuclear quantum fluctuations, as well as the richness of the physical phenomena underlying the role of NQE. Additionally, given the fact that we carefully describe the underlying nature of the mechanism as well as the physics behind the interactions, this work can serve as a solid ground and starting point to analyze NQE phenomena when different interactions are present. To the best of our knowledge this is the first time that such comprehensive and physically insightful results have been reported.

As highlighted by the reviewer one of the crucial points of our work is that the reported results were carried out using a high level of accuracy. The reason why only now we were able to carry out such extensive study is because of the predictive power and training-data efficiency of the sGDML framework, as well as its robust and efficient implementation, which allowed us to perform extensive PIMD simulations with coupled cluster accuracy. Previous studies were performed with classical force fields or sometimes with DFT, and we show that these levels of theory are often insufficient to fully capture the intricate nature of NQE effects in flexible molecules [Chmiela et al. Nat. Commun. 9, 3887 (2018); Saucedo et al. JCP 153, 124109 (2020)].

Reviewer #2 (Remarks to the Author):

The authors aim to elucidate phenomena surrounding the quantum nuclear effect (NQE) in small molecules using the sGDML machine learning-based force field along with the i-PI package to conduct classical molecular dynamics (MD) and path integral molecular dynamics (PIMD). They evaluate three phenomena that are found to be strengthened by NQE, and result in the stabilization of dynamics: the $n \rightarrow \pi^*$ interaction in Aspirin, a methyl rotor, and non-covalent interactions. Comparisons between classical molecular dynamics and path integral molecular dynamics simulations using the sGDML model provide evidence for the author's conclusions. The conclusions result in the new insight that NQE not only cause delocalization but also localization in specific cases, which could have relevance in biochemical applications.

I have no specific complaints or observations. Overall, the conclusions in the present article are sound if one trusts the underlying model used to conduct the MD and PIMD simulations. Previously published literature show the sGDML model (data set construction and model itself) can accurately fit to small data sets of small molecules. They have been shown to produce models with energy errors in the range of 0.1-0.3 kcal/mol. Data sets for the specific molecules studied in this work were generated for previous work and evaluated extensively for room temperature dynamics. Therefore, I believe the conclusions of this article to be scientifically sound.

The use of machine learning methods to obtain new insights into chemistry and physics phenomena is intriguing and of interest to a broad audience. This, in my opinion, makes the article a good selection for publication in Nature Communications.

I recommend the article for publication in Nature Communications.

Response:

We thank the referee for a careful reading of our manuscript and his/her positive comments.

Reviewer #3 (Remarks to the Author):

The manuscript of Saucedo et al. presents a thorough examination of the counter-intuitive contribution of nuclear quantum effects (NQE) on non-covalent interactions, both at intra- and intermolecular level. This unusual increase is a result of nuclear delocalization, as it was found from ab initio path integral molecular dynamics. I cannot express an opinion on this article since I am not convinced if this conclusion is based on computational artifacts or not. The authors should provide a numerical argument that the counter-intuitive contribution of NQE is not an artifact of the force fields (FFs) generated by sGDML.

Response (1):

We thank the referee for highlighting our thorough examination of the counter-intuitive contribution of nuclear quantum effects to both covalent and noncovalent interactions in molecular systems.

We agree with the referee that the results of any computational study must be robust and free from artifacts. In the particular context of our study, there are two aspects to assess the robustness of our conclusions: 1) The accuracy of the sGDML models for the relevant physical conditions of application, and 2) the accuracy of the reference data (level of electronic-structure theory and one-electron basis set for expanding the wave function) used to construct the sGDML models. Both aspects have been systematically assessed in our work and our conclusions have been fully confirmed by different levels of electronic-structure theory and basis sets expansions.

Regarding the accuracy of sGDML models, the reliability of the trained sGDML force fields for all the molecules studied in this work has been assessed in previous publications that did not focus on NQE [Chmiela et al. Nat. Commun.9, 3887 (2018); Chmiela et al. Sci. Adv. 3, e1603015 (2017); Saucedo et al. JCP 150 (11), 114102 (2019), Saucedo et al. "Construction of machine learned force fields with quantum chemical accuracy: Applications and chemical insights," in Machine Learning Meets Quantum Physics (Springer International Publishing, 2020)].

sGDML is a state-of-the-art machine learning framework for the construction of accurate and efficient molecular force fields. The reliability and applicability of sGDML force fields have been established based on rigorous statistical learning theory and by direct comparison between results obtained from the sGDML force field and electronic-structure simulations. Each one of the sGDML-molecular force fields is constructed by training its adjustable parameters on a training set and tested on a separate unseen dataset of molecular conformations. This ensures that the model is able to generalize to the broad set of molecular configurations explored by molecular dynamics simulations at finite temperatures.

sGDML

DFT

A Force Prediction

B Energy Prediction

Fig. R1. Energy and force prediction accuracy (in terms of the mean absolute error (MAE)) as a function of training set size of both models trained on DFT forces: the gain in efficiency and accuracy is directly linked to the number of symmetries in the system. Taken from Ref. Chmiela et al. Nat. Commun.9, 3887 (2018).

Fig. R1 demonstrates the reliability of sGDML force field models in terms of generalization error (error on unseen data, i.e. on data not used for training) as a function of the training set size for ten molecules, which includes all the molecules studied in this work. The sGDML models display a fast saturation of the generalization accuracy with the size of the training set. In our work, we have used models trained on 1000 training molecular configurations, clearly reaching the saturation limit. The generalization error across the potential energy surface for all the molecules included in this study is below 0.2 kcal/mol for energies and below 0.7 kcal/mol/Angstrom for forces.

The applicability of the sGDML models can be demonstrated by calculating thermodynamic observables by performing classical or path integral molecular dynamics simulations. Fig. R2 demonstrates the predictive power of the sGDML models by replicating the trajectory generated

through DFT simulations (for up to 25 picoseconds) as well as its energy distribution. Such predictive accuracy in the trajectories is corroborated by replicating the interatomic distance distribution shown in Fig. R2-right and Fig. R3.

[REDACTED]

Taken from Saucedo et al. "Construction of machine learned force fields with quantum chemical accuracy: Applications and chemical insights," in Machine Learning Meets Quantum Physics (Springer International Publishing, 2020).

[REDACTED]

Sci. Adv. 3, e1603015 (2017).

For more details refer to the original article Chmiela et al.

Fig. R4. Joint probability distribution function for the two dihedral angles of the methyl and hydroxyl functional groups in ethanol molecule. Each minimum is annotated with the occupation probability obtained from classical and path-integral MD in comparison with experimental values. Taken from Chmiela et al. *Nat. Commun.* 9, 3887 (2018).

Furthermore, in Fig. R4 sGDML path integral simulations have displayed a remarkable accuracy on replicating the experimental statistical occupations of the ethanol minima and vibrational spectrum as reported in Chmiela et al. *Nat. Commun.* 9, 3887 (2018).

To summarize, by running molecular dynamics using the sGDML force fields we were able to 1) reproduce the exact electronic structure trajectory starting from the same initial conditions (Fig. R2-left, center), 2) recover the interatomic distance distribution for several molecules relative to *ab initio* simulations (Fig. R2-right, Fig. R3), and 3) to corroborate statistical occupation of molecular conformers relative to experimental results (Fig. R4).

The different levels of generalization accuracies and applicability of the sGDML models just summarized above demonstrate the robustness and reliability of the method, discarding any potential artifacts in the simulations within the regimes of application. A compact version of this discussion referencing the appropriate work has been included in subsection A (“Machine learned force fields: sGDML”) of the Methodology section.

Will classical FFs or FFs generated by other ML methods provide the same results? This argument will further enhance their conclusions.

Response (2):

Regarding classical force fields (FF): Classical FF and ML based FFs are fundamentally different. Classical FF are based on hand-crafted (mostly harmonic) analytical formulas with parameters fitted to experimental results or theoretical calculations. Contrasting this approach, ML based FFs are general methodologies with solid foundations in statistical learning theory and no explicit analytic bias in the interatomic potential formulation. Classical force fields in general yield wrong results for NQE (see discussion below) because they are unable to model any of the electronic effects considered in our study. This can be seen in Fig. R5, where the free energies generated by classical MD simulations at 300K using sGDML@CC and classical FF (AMBER) are compared for three different molecules. The results in this figure display a clear difference between the two methodologies, as a consequence of the different underlying potential energy surface. This comparison is analyzed in detail in Chmiela et al. *Nat. Commun.* 9, 3887 (2018) and more recently in Saucedo et al. *JCP* 153, 124109 (2020). This last citation has been added to the manuscript.

Regarding alternative ML FFs: While a number of competing ML FF methodologies have been proposed in the literature, neither of them combines the learning efficiency and computational efficiency of sGDML. The generalization accuracy and applicability of sGDML presented above proves beyond any doubt that our conclusions are robust and will “stand the test of time”, as nicely remarked by Reviewer #1.

Fig. R5. Accuracy of the *sGDML* model in comparison to a traditional force field. We contrast the dihedral angle probability distributions of ethanol, malonaldehyde, and aspirin obtained from classical MD simulations at 300 K with *sGDML* (left column) vs. the *AMBER* (right column) force field. The ethanol simulations were carried out at constant energy (*NVE*), whereas a constant temperature (*NVT*) was used for malonaldehyde and aspirin. a) Ethanol: the coupling between the hydroxyl and methyl rotor is absent in *AMBER*. Moreover, the probability distribution shows an unphysical harmonic sampling at room temperature, revealing the oversimplified harmonic description of bonded interactions in that force field. b) Malonaldehyde and c) aspirin: the formulation of the *AMBER* force field is dominated by Coulomb interactions, which can lead an incomplete description of the PES and even spurious global minima in the case of aspirin. The length of the simulations was 0.5 ns. Figure taken from Chmiela et al. *Nat. Commun.* 9, 3887 (2018).

In the Methodology section, the authors mention that the *sGDML* models were trained on CCSD/cc-pVDZ, CCSD(T)/cc-pVDZ and PBE0+MDB/really_tight. I have many questions here.

1. Which quantum chemical method was used for each molecular system that is discussed on the manuscript?

Response (3):

Based on the assessed accuracy and applicability of the trained *sGDML* models in our previous publications (Chmiela et al. *Nat. Commun.* 9, 3887 (2018) and Saucedo et al. *JCP* 150 (11), 114102 (2019)), in the original manuscript we did not consider necessary to be more specific. Nevertheless, based on the comment of the referee, we have added a subsection (“Reference

molecular databases”) in the Methodology section (page 8) where we describe the methodologies used for each of the molecules in detail.

For the main results in our manuscript, aspirin was trained on CCSD calculations using the cc-pVDZ basis, and toluene and benzene were trained on CCSD(T)/cc-pVDZ data. In the case of paracetamol we used PBE0+MBD with the basis settings really_tight as implemented in the FHI-aims code (See Response (6) for more details on the DFT calculations and the basis set type).

It is important to highlight that the reference level of theory for each molecule was selected based on its accuracy relative to CCSD(T)/cc-pVDZ. In the case of aspirin, CCSD/cc-pVDZ already gives good results compared to CCSD(T)/cc-pVDZ, hence we use such approximation. For paracetamol, we have used PBE0+MBD/really_tight given that it provides better energetic values relative to CCSD(T)/cc-pVDZ than CCSD/cc-pVDZ. See the following table (which has been added to the SI).

Table 1. Energies relative to the global minimum in kcal/mol computed with different levels of theory.

Paracetamol mol.	CCSD/cc-pVDZ	CCSD(T)/cc-pVDZ	PBE0+MBD/r_t
Local minimum	0.38	0.38	0.37
Transition state (TS)	3.06	3.44	3.45
TS - Local	2.68	3.07	3.08

2. It is well-known that coupled-cluster needs very large basis sets (at least of quadruple-zeta quality or extrapolation to the complete basis set limit) in order to provide the expected accuracy. How the choice of a basis set will affect the PIMD results and, especially, the counter-intuitive phenomena? The cc-pVDZ basis set is an extremely small basis and provides inaccurate results for non-covalent interactions.

Response (4):

The referee is right only when referring to coupled cluster calculations of intermolecular interactions or large molecules. For small molecules considered in our work, the convergence of their thermodynamical and statistical properties do not require large basis sets.

To show this, Fig. R6 presents the results of classical MD and PIMD simulations using a sGDML model for aspirin trained on CCSD/cc-pVTZ reference calculations, which display the same results as the ones in Fig. 2 in the main text constructed using the cc-pVDZ basis set for computing atomic forces. This means that intramolecular interactions in small molecules are already statistically well captured with cc-pVDZ basis set.

PES sampling by MD simulations using sGDML@{CCSD/cc-pVTZ}

Fig. R6. Classical (MD) and path integral molecular dynamics (PIMD) simulations at room temperature of aspirin described by the sGDML@CCSD/cc-pVTZ molecular force field. The plots are projections of the dynamics to the two main degrees of freedom of aspirin: carboxyl and ester dihedral angles. The results in this plot are unpublished.

In the case of toluene, we have already shown evidence in the main text and in the SI that the localization of the methyl rotors is agnostic to the employed level of electronic-structure theory (see page 4 main text and Fig-SI 1 in the SI). Based on all this evidence, we conclude that our results are sufficiently converged in terms of the basis set and our conclusions regarding the role of NQE are robust.

In addition, for all molecules in our study we also constructed sGDML models with PBE0+MBD/really_tight reference data. The energies and forces of these calculations are free from any basis set artifacts and are converged to meV-level accuracy. PIMD simulations using sGDML@PBE0+MBD models yield the same conclusions as the PIMD simulations presented in the manuscript.

This discussion has been added to the subsection “Reference molecular datasets” in Methodology and to the SI.

For example, for the benzene dimer (parallel displacement), a test case that is used by the authors on this paper, the slightly larger aug-cc-pVDZ basis has an error of -1.87 and 0.63 kcal/mol for CCSD and CCSD(T) methods, respectively (see http://vergil.chemistry.gatech.edu/active_bfdb/bfdb/cgi-bin/reactionviewer.py?dataset=S22&rxn=S22-11)

Response (5):

We do not use CCSD or CCSD(T) methods to compute non-bonded interactions. Instead, to generate the data in Fig. 5 of the manuscript we have used SAPT (specifically SAPT0/jul-cc-pVDZ as mentioned in caption Fig. 5) which has been repeatedly proven to give good results in particular in the type of system studied here [Parker et al. J. Phys. Chem. 140, 094106 (2014)], as shown in the case of the relaxed energy curves in Fig. 5. Furthermore, in this work, we are interested in the relative enhancement of the interaction energy between fragments induced by the nuclear quantum effects and not only on pure binding energy values. Thereby, this level of theory is a robust approximation to demonstrate the enhancement of the interaction.

3. What means really_tight for PBE0+MDB. The PBE0+MDB does not require standard basis functions?

Response (6):

The FHI-aims package that we used for DFT calculations uses numerically tabulated atom-centered orbitals (NAOs), which are constructed to be transferable and hierarchical basis sets to systematically reach basis set convergence (sub-meV-level accuracy on the total energy) in DFT calculations. On the other hand, MBD energy corresponds to the exact diagonalization of the coupled quantum harmonic oscillator Hamiltonian in the basis set of atomic positions (hence, there is no one-particle basis set dependence in MBD). The DFT basis sets are constructed starting from a minimal basis set (minimal free-atom basis) and then systematically adding new energetically-favorable functions from a pool of basis functions (hydrogen-like, ion-like, or gaussian functions with a variable confinement potential). The basis used in this work, named “really_tight” in the code (tier 2), consists of hydrogen-like functions for C[{minimal}*]+H(*nl,z*): {He}+2s 2p]+H(4f,9.8). A more detailed description can be found in the original article by Blum et al. Comput. Phys. Commun. 180, 2175 (2009). The binding energies between small molecules with “really_tight” basis sets are converged to 1 meV-level accuracy.*

A version of this description has been added to the Supporting Information.

Similarly, how accurate are the computed polarizabilities at the CCSD/aug-cc-pVDZ level of theory?

Response (7):

The CCSD/aug-cc-pVDZ level of theory already recovers 87% of the experimental molecular polarizability, nevertheless, here we are interested in analyzing the relative gain in the polarizability by the thermal fluctuations and NQE relative to the equilibrium molecular configuration as described in the main text. Hence, the theoretical analysis presented in section Results: “Impact of NQE on van der Waals interactions” along with this level of theory provide a robust approximation for this purpose.

This discussion has been added to the main text in section Results / Impact of NQE on van der Waals interactions.

Caption of Figure 3 (SI), please change “inividual” to “individual”.

Response (8):

We thank the reviewer for pointing out this oversight. This has been corrected.

REVIEWERS' COMMENTS

Reviewer #3 (Remarks to the Author):

The authors have provided adequate information (previous literature, numerical data) to justify that their conclusions for NQEs are not artifacts. Both manuscript and SI have been updated. I am still though not fully convinced about the accuracy of CCSD/CCSD(T) data computed with a double-zeta basis, but probably the expected agreement is due to cancelation of errors. I do not have any further comments.

Response to the reviews:

Reviewer #3 (Remarks to the Author):

The authors have provided adequate information (previous literature, numerical data) to justify that their conclusions for NQEs are not artifacts. Both manuscript and SI have been updated. I am still though not fully convinced about the accuracy of CCSD/CCSD(T) data computed with a double-zeta basis, but probably the expected agreement is due to cancelation of errors. I do not have any further comments.

Response:

We thank the referee for a careful reading of our article and her/his positive feedback.